# Surgical decision making in the setting of severe traumatic brain injury: A survey of neurosurgeons

Theresa Williamson[1]*, Marc D. Ryser[2,3,4], Jihad Abdelgadir[1], Monica Lemmon[5,6], Mary Carol Barks[7], Rasheedat Zakare[3], Peter A. Ubel[3,7]

1 Department of Neurosurgery, Duke University Medical Center, Durham, North Carolina, United States of America, 2 Department of Population Health Sciences, Duke University Medical Center, Durham, North Carolina, United States of America, 3 Duke School of Medicine, Duke University, Durham, North Carolina, United States of America, 4 Department of Mathematics, Duke University, Durham, North Carolina, United States of America, 5 Department of Pediatrics, Duke University Medical Center, Durham, North Carolina, United States of America, 6 Duke-Margolis Center for Health Policy, Durham, North Carolina, United States of America, 7 The Fuqua School of Business, Duke University, Durham, North Carolina, United States of America

* theresa.williamson@duke.edu

**Data Availability Statement:** All relevant data are within the manuscript and its supporting information files.

## Abstract

### Background

Surgical decision-making in severe traumatic brain injury (TBI) is complex. Neurosurgeons weigh risks and benefits of interventions that have the potential to both maximize the chance of recovery and prolong suffering. Inaccurate prognostication can lead to over- or under-estimation of outcomes and influence treatment recommendations.

### Objective

To evaluate the impact of evidence-based risk estimates on neurosurgeon treatment recommendations and prognostic beliefs in severe TBI.

### Methods

In a survey-based randomized experiment, a total of 139 neurosurgeons were presented with two hypothetical patient with severe TBI and subdural hematoma; the intervention group received additional evidence-based risk estimates for each patient. The main outcome was neurosurgeon treatment recommendation of non-surgical management. Secondary outcomes included prediction of functional recovery at six months.

### Results

In the first patient scenario, 22% of neurosurgeons recommended non-surgical management and provision of evidence-based risk estimates increased the propensity to recommend non-surgical treatment (odds ratio [OR]: 2.81, 95% CI: 1.21–6.98; p = 0.02). Neurosurgeon prognostic beliefs of 6-month functional recovery were variable in both control (median 20%, IQR: 10%-40%) and intervention (30% IQR: 10%-50%) groups and

**Funding:** The author(s) received no specific funding for this work.

**Competing interests:** The authors have declared that no competing interests exist.

neurosurgeons were less likely to recommend non-surgical management when they believed prognosis was favorable (odds ratio [OR] per percentage point increase in 6-month functional recovery: 0.97, 95% confidence interval [CI]: 0.95–0.99). The results for the second patient scenario were qualitatively similar.

## Conclusions

Our findings show that the provision of evidence-based risk predictions can influence neurosurgeon treatment recommendations and prognostication, but the effect is modest and there remains large variability in neurosurgeon prognostication.

## Introduction

Severe traumatic brain injury (TBI) accounts for 52,000 patient deaths and 5.3 million newly disabled Americans yearly[1–4]. For patients diagnosed with severe TBI with subdural hematoma, there are three dominant treatment modalities: craniotomy, non-surgical medical treatment and withdrawal or withholding of treatment. TBI patients are usually in critical condition and treatment decisions are time-sensitive. Physician recommendations play a critical role in decision-making because surrogate decision makers are faced with an overwhelming and unfamiliar scenario. When making the treatment recommendation, neurosurgeons, emergency, and neurocritical care providers face a critical trade-off: on the one hand, performing a craniotomy can relieve pressure from subdural hematoma and thus increase the chance of recovery and minimize the risk of neurologic decline; on the other hand, there is a non-negligible risk, especially in elderly patients, that the invasive procedure may prolong suffering without benefit[5, 6]. Making this decision even more difficult, there is large uncertainty about which patients will benefit from craniotomy in the setting of severe TBI and uncertainty about TBI outcomes[7, 8]. According to a position statement by the Neurocritical Care Society, there is concern that uncertain neuroprognostication limits treatment decisions[9]. Indeed, a previous study found that neurosurgeons tend to overestimate poor outcomes and underestimate positive outcomes in the setting of severe TBI[10].

Over the past years, there has been an increased effort to derive evidence-based treatment recommendations for patients with TBI. Several studies have used cohort and registry data to develop and validate quantitative risk prediction models based on clinical features[11, 12]. Furthermore, through the development of web-based risk calculators these evidence-based outcome predictions are now readily accessible to clinicians. Two such risk calculators are CRASH[11] and IMPACT[12], which both incorporate patient presentation data to predict multiple short and long-term outcomes[13, 14]. Although validated, these calculators are not widely used in clinical practice[15]. Therefore, little is known about the effect of prognostic calculators on neurosurgeon prognostic beliefs and treatment recommendations.

The objective of this study was two-fold: 1) to characterize neurosurgeon decision making in severe TBI and 2) to evaluate the impact of evidence-based prognostic risk estimates on neurosurgeon prognostic beliefs and treatment decisions. To this end, we performed a survey of neurosurgeons at an international meeting and assessed their beliefs and recommendations in hypothetical scenarios of patients with severe TBI. The survey included a randomized experiment to evaluate the effect of providing evidence-based prognostic risk estimates.

## Methods

### Survey design

We surveyed a convenience sample of neurosurgeons attending the Congress of Neurological Surgeons Annual Meeting in 2017 to determine their prognostic estimates and treatment recommendations for hypothetical patients with severe traumatic brain injury. Given their integral involvement in neurosurgical decision-making, attending, fellow, and resident level neurosurgeons were eligible for the study. The study was open to neurosurgeons from any country. Physicians from other specialties and medical students were excluded. Ethics approval for this study was obtained from the DukeHealth Institutional Review Board. The survey contained a statement informing participants of the goals and details of the study and giving an opportunity for consenting to or declining participation.

The control survey contained two hypothetical patient scenarios and no prognostic estimate information (Table 1 and Supplemental Digital Content): one patient was a 77-year-old patient, Glasgow Coma Scale Total Score five, with reactive pupils (Scenario one); and a 60-year-old patient, Glasgow Coma Total Score four with fixed and dilated pupils (Scenario two). In both scenarios, patients were described as having one centimeter subdural hematomas on computed tomography (CT) scan (laterality unspecified) with one centimeter of midline shift. The order of presentation of scenarios was the same on all surveys. The intervention survey contained the same hypothetical patient scenarios as the control survey, and additionally provided prognostic estimates based on the CRASH calculator[14]. Prognostic estimates in the CRASH survey were comprised of the risk of 14-day mortality and the risk of 6-month poor functional outcome, defined as death, vegetative state, or severe disability[14]. For scenario one, the CRASH calculator estimated a 14-day mortality of 65.6% and a 6-month risk of poor functional outcome of 93%; for scenario two, the CRASH calculator predicted a 14-day mortality of 73.1% and a 6-month risk of poor functional outcome of 93%. Upon reading the patient scenarios, the survey respondents were first asked to estimate the percent chance of 30-day survival and the percent chance that at six months the patient would be able to communicate and perform activities of daily living (6-month ADL). Next, respondents were asked to answer a hypothetical family member's question about treatment recommendation with the following choices: craniotomy, medical management, or comfort care, as defined as the decision to withdraw or withhold life-sustaining intervention.

In addition to the scenario responses, participants were asked to provide the following demographic information: age, experience (defined as the time since graduation from medical

**Table 1. Scenario descriptions.** Hypothetical patient characteristics presented to neurosurgeon participants.

|  | Patient 1 | Patient 2 |
|---|---|---|
| Age (years) | 77 | 60 |
| Mechanism of Injury | Fall | |
| Time Since Injury | Unknown | |
| Comorbidities | Unknown | |
| GCS | 5 | 4 |
| CT brain | 1cm acute subdural hematoma, 1cm midline shift, scattered traumatic subarachnoid hemorrhage | |
| Labs | Glucose 90mg/dL, coagulation within normal limits, hemoglobin 9g/dL | |
| CRASH estimate of 14-day mortality risk | 65.60% | 73.10% |
| CRASH estimate of 6-month unfavorable outcome risk | 93.00% | |

school), gender, race/ethnicity, training level (intern, resident, fellow or attending), and hospital type (level 1 trauma center: yes/no).

## Sample size

We hypothesized that respondents receiving the intervention survey including evidence-based risk estimates would both predict poorer prognoses and be less likely to recommend craniotomy than respondents receiving the control survey. The experiment was powered on the effect of evidence-based risk estimates on craniotomy recommendation in the first scenario. Based on a single-institution pilot study of 50 neurosurgeons, we estimated that we needed 200 respondents (100 per group) to have 80% power to detect a medium effect size (Cohen's D = 0.4) for the impact of CRASH estimates on craniotomy recommendations, assuming a type I error rate of 0.05.

## Statistical analyses

Survey responses were double-entered and discrepancies were resolved by referring back to the written survey. Surveys were considered complete if at least 50% of the questions were answered; incomplete surveys and surveys with missing treatment recommendation were excluded from the analysis. Continuous prognostic predictions between the two study groups were compared using the two-sided t-test. The association between survey version and recommendation of non-surgical management was estimated using univariable logistic regression models and odds ratios (OR) were calculated. The association between survey version and the three management options of craniotomy, medical management and comfort care was estimated using multinomial logistic regression models, and corresponding ORs were calculated. For exploratory mediation analyses[16], the effect of neurosurgeon prognostic beliefs on treatment recommendation was evaluated using univariable and multivariable logistic regression models, the latter adjusted for neurosurgeon age, race, gender, experience, training level, hospital type and survey version. All statistical tests performed were two-sided. Analyses were performed in STATA (version 15.0.587, StataCorp. 2017, College Station, TX) and R (version 3.5.1, R Foundation for Statistical Computing, Vienna, Austria).

## Results

### Demographics

A total of 139 neurosurgeons completed the survey. After removal of surveys with incomplete treatment recommendations, 138 and 136 surveys were included for the analyses of scenarios one and two, respectively. Among the 138 respondents of scenario one (Table 2), the majority were male (80%), and median age was 40 (interquartile range [IQR]: 30–55) years. The majority of respondents were either fellows or attending surgeons (65%), and median professional experience was 13 (IQR: 5–27) years. A total of 68 (49%) participants received the control version of the survey without CRASH estimates and 70 (51%) received the intervention version with the CRASH estimates.

### Prognostic beliefs

Overall, neurosurgeon predictions of 30-day survival and 6-month ADL were widely variable regardless of survey version received (Fig 1). In scenario one (Fig 1A), the median 30-day survival prediction among neurosurgeons who received evidence-based risk estimates was 40% (IQR: 30%-68%), compared to 58% (IQR: 30%-80%) among those who did not receive the risk estimates. For 6-month ADL, the median prediction among neurosurgeons who received risk

**Table 2. Survey participant demographics for scenario 1.**

|  | All participants (n = 138*) | Control (n = 68*) | Intervention (n = 70*) |
|---|---|---|---|
| **Age [y], median (IQR)** | 40 (30–55) | 40 (32–53) | 40 (33–57) |
| **Experience [y], median (IQR)** | 13 (5–27) | 13.5 (2–26) | 13 (6–31) |
| **Gender, n (%)** |  |  |  |
| Male | 110 (80) | 52 (77) | 58 (83) |
| Female | 26 (19) | 15 (22) | 11 (16) |
| Unknown | 2 (1) | 1 (1) | 1 (1) |
| **Race/Ethnicity, n (%)** |  |  |  |
| White | 78 (57) | 38 (56) | 40 (57) |
| Black | 11 (8) | 3 (4) | 8 (11) |
| Asian | 24 (17) | 14 (21) | 10 (14) |
| Hispanic | 13 (9) | 7 (10) | 6 (9) |
| Unknown | 12 (9) | 6 (9) | 6 (9) |
| **Training level, n (%)** |  |  |  |
| Resident | 45 (33) | 23 (34) | 22 (32) |
| Attending/Fellow | 90 (65) | 43 (63) | 47 (67) |
| Unknown | 3 (2) | 2 (3) | 1 (1) |
| **Hospital type, n (%)** |  |  |  |
| Level 1 trauma center | 91 (66) | 43 (32) | 22 (31) |
| Other | 44 (32) | 22 (63) | 48 (69) |
| Unknown | 3 (2) | 3 (4) | 0 (0) |

*For the analysis of scenario 2, n = 2 participants were excluded due to missing treatment recommendation.

estimates was 20% (IQR: 10%-40%), compared to 30% (IQR: 10%-50%) among those who did not receive estimates.

In scenario two (Fig 1B), the median 30-day survival prediction was lower among neurosurgeons who received risk estimates (28%, IQR: 15%-50%) compared to those who did not (40%, IQR: 10%-60%). Similarly, the median 6-month ADL prediction among those who received risk estimates (6%, IQR: 1%-24%) was lower compared to those who did not (10%, IQR: 1%-50%).

## Treatment recommendations

In both survey groups, neurosurgeons recommended non-surgical treatment (including medical management or withdrawal of life sustaining treatment) over a wide range of prognostic beliefs (Fig 1, black dots). Overall, 22% and 42% of neurosurgeons recommended non-surgical treatment in scenarios one and two, respectively (Table 3). In scenario one, neurosurgeons who received risk estimates were more likely to recommend non-surgical treatment (OR: 2.81, 95% CI: 1.21–6.98; p = 0.02). In scenario two, there was no difference in non-surgical management choices between the control and intervention groups (OR: 1.37, 95% CI: 0.69–2.74). When further stratifying the non-surgical management recommendations into medical treatment and withdrawal of life-sustaining intervention in scenario one, we found that neurosurgeons receiving risk estimates were more likely to choose withdrawal (OR: 5.42, 95% CI: 1.12–26.25), but not medical treatment (OR: 2.06, 95% CI: 0.75–5.65). No effect was found in scenario two.

**Exploratory mediation analysis for scenario one.** In scenario one, the provision of evidence-based risk estimates had an effect on both 6-month functional recovery beliefs (p = 0.06, t-test) and treatment recommendations (p = 0.02, Table 3). We thus performed an

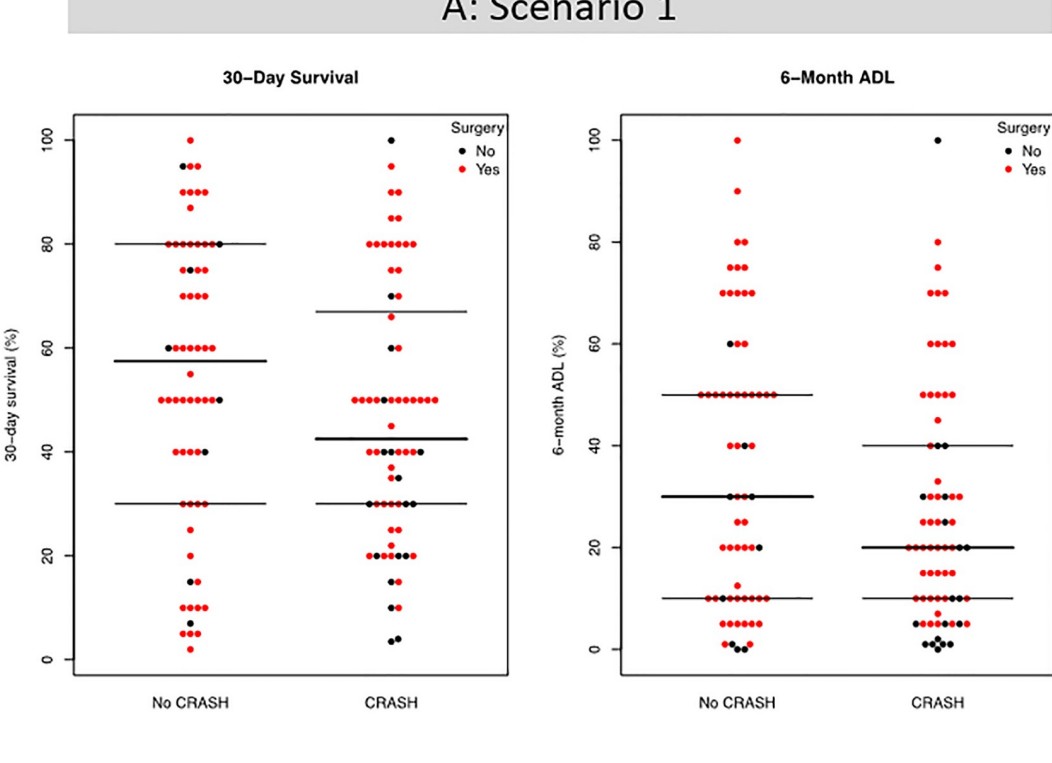

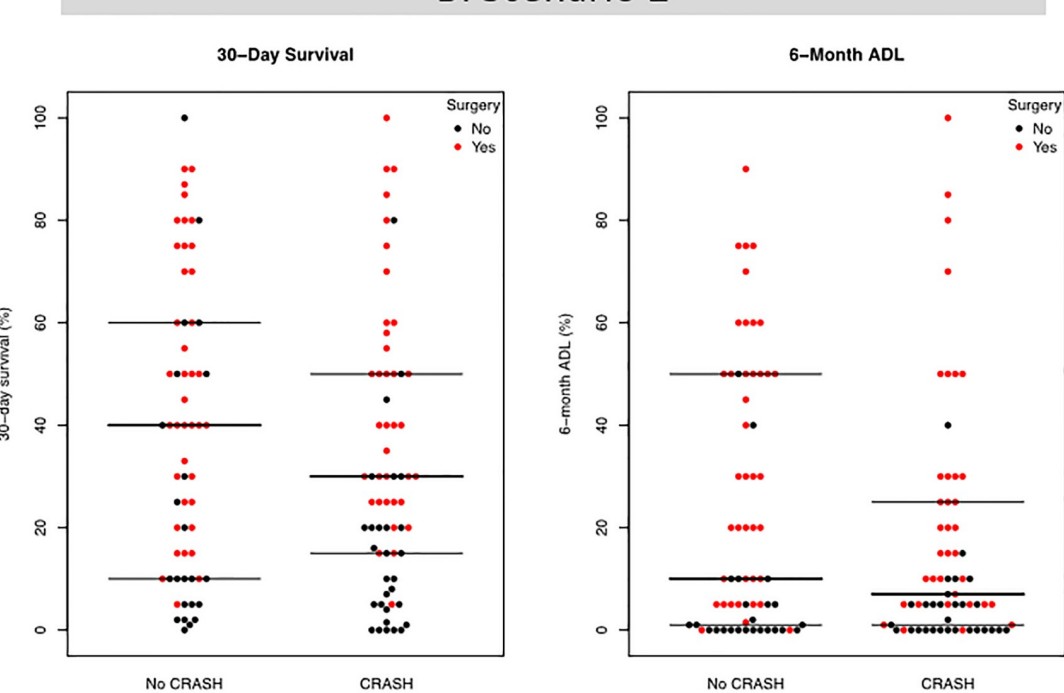

**Fig 1. Neurosurgeon prognostic beliefs.** Neurosurgeon prognostic beliefs about 30-day survival and 6-month ADL are shown for the hypothetical scenarios 1 (A) and 2 (B). Each dot represents an individual neurosurgeon prognostic estimate. Red dots represent neurosurgeons who recommended a craniotomy for the hypothetical patient and black dots represent neurosurgeons who did not recommend a craniotomy. Horizontal lines represent the median (middle line) and interquartile range (top and bottom lines) of the predictions, respectively.

**Table 3. Effect of providing risk estimates on treatment choice.**

| | | Scenario One | | | Scenario Two | |
|---|---|---|---|---|---|---|
| | Control (N = 68) | Intervention (N = 70) | Odds ratio (95% CI) | Control (N = 67) | Intervention (N = 69) | |
| **Odds ratio (95% CI)** | | | | | | |
| **n (%)** | | | | | | |
| **Non-surgical management** | 9 (13) | 21 (30) | 2.8 (1.2–7.0)* | 25 (37) | 31 (45) | 1.4 |
| (0.7–2.7) | | | | | | |
| **Non-surgical management, by type** | | | | | | |
| Medical management | 7 (10) | 12 (17) | 2.1 (0.8–5.6) | 11 (16) | 9 (13) | 0.9 |
| (0.3–2.4) | | | | | | |
| Comfort care | 2 (3) | 9 (13) | 5.4 (1.1–26.2) | 14 (21) | 22 (32) | 1.7 |
| (0.8–3.9) | | | | | | |

*p = 0.02

exploratory mediation analysis to evaluate the potential role of 6-month ADL as a mediator for the effect of risk estimate provision on treatment recommendation (Fig 2). In a univariable analysis of 6-month ADL as a predictor for treatment recommendation, there was a negative association between 6-month ADL estimates and non-surgical treatment recommendation (OR per percentage point increase in 6-month ADL: 0.97, 95% CI: 0.95–0.99, p = 0.01; Table 4). When adjusting the analysis for neurosurgeon characteristics and survey version, the

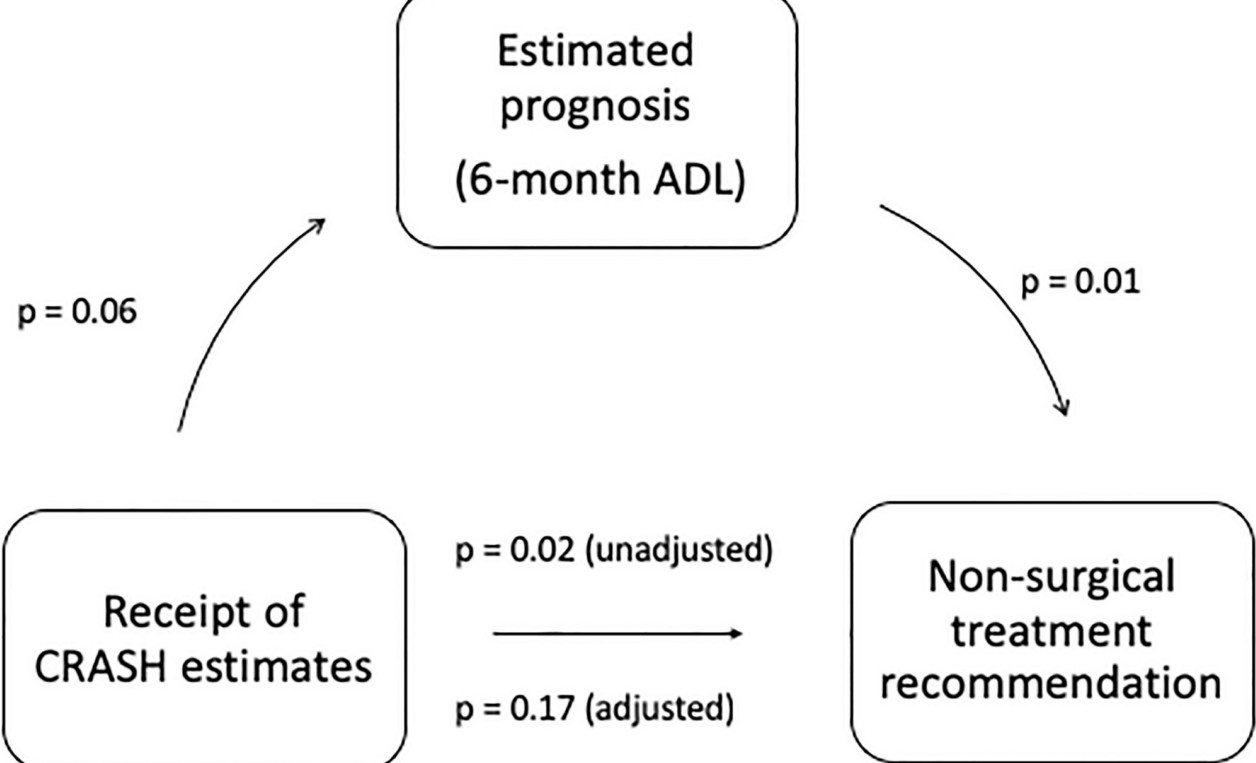

**Fig 2. Exploratory mediation analysis.** Relationship between receipt of evidence-based risk estimates, prognostic beliefs and treatment recommendation. When including prognostic estimates in the model, the relationship between receipt of evidence-based risk estimates and treatment recommendation loses statistical significance.

**Table 4. Effect of prognostic beliefs (6-month ADL) on recommendation of non-surgical management.**

| 6-Month ADL | | | |
|---|---|---|---|
| **Analysis** | **Variable** | **Odds ratio (95% CI)** | **p-value** |
| Univariable | 6-month ADL | 0.97 (0.95–0.99) | 0.01 |
| Multivariable adjusted | 6-month ADL (percentage points) | 0.97 (0.94–0.99) | 0.01 |
| | Gender: male | 1.49 (0.44–5.86) | 0.54 |
| | Race: white | 0.34 (0.12–0.90) | 0.03 |
| | Age (years) | 1.04 (0.88–1.20) | 0.62 |
| | Hospital type: Level 1 trauma center | 0.95 (0.27–3.44) | 0.94 |
| | Position: intern/resident | 0.68 (0.18–2.55) | 0.57 |
| | Experience (years) | 0.93 (0.81–1.09) | 0.34 |
| | Version: CRASH | 2.03 (0.76–5.84) | 0.17 |

strength of the association between 6-month ADL and non-surgical recommendation remained unchanged. Importantly, in the multivariable analysis, the survey version was no longer a predictor for the treatment recommendation (Table 4). Together, these results suggest that prognostic beliefs (6-month ADL) may be a potential mediator in the association of survey version and treatment recommendation. In scenario two, a mediation analysis was not warranted because provision of evidence-based risk estimates had no effect on treatment recommendation.

## Discussion

We conducted a survey to evaluate the role of evidence-based risk predictions in neurosurgeon prognostication and decision-making in severe TBI. Our study yielded three main insights. First, neurosurgeon prognostic predictions were highly variable and only modestly influenced by the provision of evidence-based risk predictions. Second, the majority of neurosurgeons recommended craniotomy for hematoma evacuation, although they were less likely to do so when they believed the prognosis was very poor. Third, the provision of evidence-based risk predictions decreased the propensity to recommend craniotomy, and this effect may have been partially mediated by a change in neurosurgeon belief about prognosis.

In both scenarios, there was very little prognostic agreement among neurosurgeons, an observation that applied to both the control and intervention groups of the study. This variability in prognostication and decision making in severe TBI may reflect tensions in the existing prognostic literature. On one hand, there is rising evidence of delayed neurologic recovery in patients previously believed to have devastating prognoses[17, 18]. On the other hand, there is growing evidence of over-utilization of surgery at the end of life[6], limited cost-effectiveness of craniotomy for TBI in patients with very poor prognoses[19], and the fear of leaving patients in a state they would find unacceptable[20]. It should be noted that both hypothetical patients had poor prognoses according to the CRASH calculator. Although these scenarios had similar CRASH estimates for prognosis, the patient presentations varied in ways that led neurosurgeons to different conclusions which is an interesting area of future study. Finally, the available data about the patient was limited purposefully as many of these decisions are made quickly with little clinical information in an emergent setting.

Overall, provision of evidence-based risk estimates had a limited impact on neurosurgeon prognostic beliefs. In the first scenario, there was a modest negative influence of CRASH estimates on 6-month ADL prognosis. There was no such effect of risk estimate provision on prognostic beliefs in the second scenario, which may partially be due to the overall poorer perceived prognosis for this hypothetical patient. The limited effect of CRASH estimates may be

due to the fact that neurosurgeons are not confident in the validity of the tool, that they have fixed beliefs about prognosis, or that they primarily rely on personal experience and training to make decisions. Indeed, a recent survey of neurosurgeons found that the use of risk calculators is rare in practice, and that neurosurgeons are unlikely to rely on them for decision making[15]. Finally, it is possible that neurosurgeons depending on practice type, experience, or health system in which they practice could have different responses to evidence-based risk estimates. This study included a diverse population of neurosurgeons but was not designed to delineate these differences. This represents an interesting area of future study.

Neurosurgeons who received evidence-based risk estimates were less likely to recommend craniotomy and more likely to recommend withdrawal of life-sustaining intervention. However, the majority of neurosurgeons in both scenarios, regardless of receiving the evidence-based risk estimates, recommended craniotomy. Although the effect of risk predictions was limited, exploratory analyses suggested that the impact of CRASH estimates on treatment recommendations may in part be mediated by beliefs about long-term outcomes such as functional recovery.

In the first patient scenario, belief about the patient's prognosis was associated with the decision to offer or not offer a craniotomy. In a previous study, neurosurgeons stated they would consider not offering aggressive treatment when the chance of specific poor prognoses was 80% or greater[15]. Interestingly, in our study, although poor prognosis correlated with fewer craniotomies, many neurosurgeons were still willing to recommend craniotomy at a very low chance of functional recovery. This raises important questions about the potential of prognostic models like CRASH and IMPACT to influence neurosurgeon decision-making in practice and provides an interesting avenue for future work that seeks to understand the mechanisms of neurosurgeon decision making. Additionally, future work is required to assess the relationship between emergent decision making and recent data and recommendations for delaying prognostication in severe TBI[21].

This study suggests that neurosurgeon decision making in severe TBI is highly variable, even in the presence of evidence-based prognostic estimates. Because prognostic estimates do modestly change prognostic belief, future work should address improving these estimates as well as exploring barriers to the use of prognostic prediction models by clinicians. In doing so, they must acknowledge that prognostication in severe TBI may be limited by self-fulfilling prophecy and inappropriately early decisions[21]. Additionally, while risk-based estimates provide insight into decision-making in severe TBI, there are other important factors to address such as physician communication of risk, bias, and consideration of uncertainty and timing of decisions. Because prognostication data and decision making in severe TBI are complex, there is also need for future work to characterize surgeon attitudes about the relationship between prognosis and treatment recommendation.

## Limitations

Our study has several important limitations. First, we studied a non-representative convenience sample, and the sample size was less than projected (139 instead of 200). A limited sample size could decrease the significance of our findings. However this study represents a larger than typical sample of neurosurgeons and identifies trends that are interesting for further study. Second, the survey results reflect clinician perception in hypothetical scenarios and it is possible that physicians may respond differently in comparable real-world situations[22]. Not all neurosurgeons are trained alike and although we looked at hospital type and training level, a future study could look at the effects of training and demographics on surgical decision making in TBI. Third, an order effect could have influenced responses to the second scenario

based on responses to the first scenario[23]. Finally, in neurosurgical practice, family preferences and communication play an important role in the shared decision making process. Our study was not designed to evaluate the role of physician-patient interaction in the decision making process and this should be addressed in future work. Finally, this work focused on neurosurgeons; decision making practices of other relevant specialists, including neurologists and neurointensivists, is an important area of future study.

## Conclusion

Neurosurgeons who received evidence-based risk estimates prior to making a treatment recommendation were less likely to recommend craniotomy and more likely to suggest withholding aggressive treatment. Prognostic beliefs for the same hypothetical patients were highly variable, and many neurosurgeons recommended craniotomy even if their own prognostic estimates were poor. The provision of risk estimates had limited effect on prognostic beliefs, but nevertheless, risk estimates may partially mediate the neurosurgeons' decision to recommend craniotomy, meaning that evidence-based estimates have the potential to influence neurosurgeon decision making. These findings raise important questions about if and how prognostic tools can be best incorporated into decision making in severe TBI.

## Supporting information

**S1 Survey. Survey provided to neurosurgeons.**
(DOCX)

## Author Contributions

**Conceptualization:** Theresa Williamson, Marc D. Ryser, Jihad Abdelgadir, Monica Lemmon, Mary Carol Barks, Rasheedat Zakare, Peter A. Ubel.

**Data curation:** Theresa Williamson, Marc D. Ryser, Mary Carol Barks, Rasheedat Zakare.

**Formal analysis:** Theresa Williamson, Marc D. Ryser, Jihad Abdelgadir, Monica Lemmon, Peter A. Ubel.

**Investigation:** Theresa Williamson, Mary Carol Barks, Rasheedat Zakare, Peter A. Ubel.

**Methodology:** Theresa Williamson, Marc D. Ryser, Jihad Abdelgadir, Monica Lemmon, Rasheedat Zakare, Peter A. Ubel.

**Project administration:** Theresa Williamson, Jihad Abdelgadir, Mary Carol Barks, Peter A. Ubel.

**Resources:** Jihad Abdelgadir, Mary Carol Barks, Peter A. Ubel.

**Supervision:** Theresa Williamson, Peter A. Ubel.

**Validation:** Theresa Williamson, Marc D. Ryser, Peter A. Ubel.

**Visualization:** Theresa Williamson, Jihad Abdelgadir, Monica Lemmon, Peter A. Ubel.

**Writing – original draft:** Theresa Williamson, Marc D. Ryser, Jihad Abdelgadir, Monica Lemmon, Mary Carol Barks, Rasheedat Zakare, Peter A. Ubel.

**Writing – review & editing:** Theresa Williamson, Marc D. Ryser, Mary Carol Barks, Peter A. Ubel.

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
