## [Decision Letter · Decision Letter 0]

18 Nov 2019

PONE-D-19-24519

Surgical Decision Making in the Setting of Severe Traumatic Brain Injury: A Survey of Neurosurgeons

PLOS ONE

Dear Dr. williamson,

Thank you for submitting your manuscript to PLOS ONE. After careful consideration, we feel that it has merit but does not fully meet PLOS ONE’s publication criteria as it currently stands. Therefore, we invite you to submit a revised version of the manuscript that addresses the points raised during the review process.

We would appreciate receiving your revised manuscript by Jan 02 2020 11:59PM. To enhance the reproducibility of your results, we recommend that if applicable you deposit your laboratory protocols in protocols.io, where a protocol can be assigned its own identifier (DOI) such that it can be cited independently in the future. For instructions see: http://journals.plos.org/plosone/s/submission-guidelines#loc-laboratory-protocols

We look forward to receiving your revised manuscript.

Kind regards,

Panagiotis Kerezoudis, M.D., M.S.

Academic Editor

PLOS ONE

Journal Requirements:

"Ethics approval for this study was obtained from the university institutional review board. The survey contained a statement informing participants of the goals and details of the study and giving an opportunity for consenting to or declining participation."

i) Please amend your current ethics statement to include the full name of the ethics committee/institutional review board(s) that approved your specific study.

ii) Once you have amended this/these statement(s) in the Methods section of the manuscript, please add the same text to the “Ethics Statement” field of the submission form (via “Edit Submission”).

"No".

Please provide an amended Funding Statement that declares *all* the funding or sources of support received during this specific study (whether external or internal to your organization) as detailed online in our guide for authors at http://journals.plos.org/plosone/s/submit-now.  

Please state what role the funders took in the study.  If any authors received a salary from any of your funders, please state which authors and which funder. If the funders had no role, please state: "The funders had no role in study design, data collection and analysis, decision to publish, or preparation of the manuscript."

"No".

i) Please complete your Competing Interests on the online submission form to state any Competing Interests. If you have no competing interests, please state "The authors have declared that no competing interests exist.", as detailed online in our guide for authors at http://journals.plos.org/plosone/s/submit-now

ii)  This information should be included in your cover letter; we will change the online submission form on your behalf.

6. Please amend your list of authors on the manuscript to ensure that each author is linked to an affiliation. Authors’ affiliations should reflect the institution where the work was done (if authors moved subsequently, you can also list the new affiliation stating “current affiliation:….” as necessary).

7. ** Please include your tables as part of your main manuscript and remove the individual files **. Please note that supplementary tables (should remain/ be uploaded) as separate "supporting information" files

Reviewers' comments:

Reviewer's Responses to Questions

**Comments to the Author**

1. Is the manuscript technically sound, and do the data support the conclusions?

Reviewer #1: Partly

Reviewer #2: Yes

2. Has the statistical analysis been performed appropriately and rigorously? 

Reviewer #1: No

Reviewer #2: Yes

3. Have the authors made all data underlying the findings in their manuscript fully available?

Reviewer #1: Yes

Reviewer #2: No

4. Is the manuscript presented in an intelligible fashion and written in standard English?

Reviewer #1: Yes

Reviewer #2: Yes

5. Review Comments to the Author

Reviewer #1: Overall

The two scenarios are essentially treated as two separate questions for much of the paper, which brings up the point of multiple comparisons. Either the second scenario is a “control” which the authors did not anticipate would vary with providing prognostic data, or the issue of multiple comparisons needs to be addressed/taken into account appropriately during analyses for significance. Alternatively, the authors could have performed analyses while somehow removing/regressing out the effect of the patient scenario to see the global effect across the entire study. Either way, the possibility of performing multiple comparisons needs to be addressed.

Abstract

Methods – Please more clearly explain the groups in methods, else the results section is less clear to the reader.

Results – As written it is not clear that the surgery recommendation changing is only for one of the two groups.

Paper

194 – The effect was borderline and not a significant effect, therefore one can claim it at most as a trend only.

195 – Is the mediation analysis for both scenarios or just the 1st ?

203 – The results of a multivariable analysis are less convincing when independent variable A (6-month ADL) is relatively correlated with variable D (whether the participant received CRASH data). There seems to be a decent degree of multicollinearity between 6-month ADL and test version, which depending on the strength may be an argument against performing this particular analysis at all. Did the authors validate the use of multiple linear regression by testing Variance Inflation Factor values?

Figures and Tables:

Table 2 – In the trauma center hospital section the number and percent data don’t make sense. I think the percent data in row “other” for control and intervention columns should be in the “Level 1” row AND the number data for intervention “other” should be in the level 1 row.

Reviewer #2: The authors present a survey analysis from a random convenience sample of 139 neurosurgeons from the Congress of Neurological Surgeons (CNS) meeting 2017. The goal of the study was to evaluate surgical decision making in the setting of severe TBI and to determine the impact of availability of evidence-based risk estimates on prognostication and treatment recommendation. The authors found that prognostic beliefs and surgical recommendation were highly variable and could be modestly influenced by evidence-based risk estimates. The manuscript is well-written and relevant findings appropriately discussed. I have the following comments:

1. I would advise the authors to avoid beginning sentences in the results section of the abstract with a number.

2. Although neurotrauma training is pretty standard across most residency programs, did the authors collect information about the type of practice for attending neurosurgeons? Were there differences based on type of fellowship training? Were spine surgeons more or less likely to recommend craniotomy? Did having additional neuro-critical care fellowship training influence treatment recommendation? If this information was not collected, the authors should mention the possibility of residual confounding due to such factors as one of the limitations of the results.

3. In the multivariable analysis presented in Table 4, I notice the authors did not adjust for other collected demographic characteristics like surgeon race/ethnicity, age and years of experience. Can the authors comment on that or provide a revised analysis adjusted for these covariates?

6. PLOS authors have the option to publish the peer review history of their article (what does this mean?). If published, this will include your full peer review and any attached files.

Reviewer #1: No

Reviewer #2: No

---

## [Author Response · Author response to Decision Letter 0]

21 Jan 2020

We thank the editors and reviewers for their thorough review and for providing constructive critique on the manuscript. In response to your thoughtful review, we carefully re-analyzed our paper including our methods and discussion sections to make our thought process and data clearer. We feel that the reviews have significantly improved our manuscript and look forward to your comments going forward.

---

## [Editor Report · Decision Letter 1]

28 Jan 2020

Surgical Decision Making in the Setting of Severe Traumatic Brain Injury: A Survey of Neurosurgeons

PONE-D-19-24519R1

Dear Dr. williamson,

We are pleased to inform you that your manuscript has been judged scientifically suitable for publication and will be formally accepted for publication once it complies with all outstanding technical requirements.

With kind regards,

Panagiotis Kerezoudis, M.D., M.S.

Academic Editor

PLOS ONE

Additional Editor Comments (optional):

Reviewers' comments:

The authors have addressed all of the reviewers' comments and the manuscript is now ready for publication.

---

## [Editor Report · Acceptance letter]

18 Feb 2020

PONE-D-19-24519R1 

Surgical Decision Making in the Setting of Severe Traumatic Brain Injury: A Survey of Neurosurgeons 

Dear Dr. Williamson:

I am pleased to inform you that your manuscript has been deemed suitable for publication in PLOS ONE. Congratulations! Your manuscript is now with our production department. 

With kind regards,

on behalf of

Dr. Panagiotis Kerezoudis 

Academic Editor

PLOS ONE